# Prescribing quality in secondary care patients with different stages of chronic kidney disease: a retrospective study in the Netherlands

Kirsten PJ Smits,[1] Grigory Sidorenkov,[1] Frans J van Ittersum,[2] Femke Waanders,[3] Henk JG Bilo,[4] Gerjan J Navis,[5] Petra Denig[1]

► Additional material is published online only. To view please visit the journal online (http://dx.doi.org/10.1136/bmjopen-2018-025784).

[1]Department of Clinical Pharmacy and Pharmacology, University Medical Center Groningen, Groningen, The Netherlands
[2]Department of Nephrology, VU University Medical Center Amsterdam, Amsterdam, The Netherlands
[3]Department of Nephrology, Isala Clinics, Zwolle, The Netherlands
[4]Diabetes Centre, Isala Clinics, Zwolle, The Netherlands
[5]Department of Nephrology, University Medical Center Groningen, Groningen, The Netherlands

**Correspondence to**
Dr. Kirsten PJ Smits;
kirsten.smits@radboudumc.nl,
k.p.j.smits@umcg.nl

## ABSTRACT

**Objectives** Insight in the prescribing quality for patients with chronic kidney disease (CKD) in secondary care is limited. The aim of this study is to assess the prescribing quality in secondary care patients with CKD stages 3–5 and possible differences in quality between CKD stages.

**Design** This was a retrospective cohort study.

**Setting** Data were collected at two university (n=569 and n=845) and one non-university nephrology outpatient clinic (n=1718) in the Netherlands.

**Participants** Between March 2015 and August 2016, data were collected from patients with stages 3a–5 CKD seen at the clinics. Blood pressure measurements, laboratory measurements and prescription data were extracted from medical records. For each prescribing quality indicator, patients with incomplete data required for calculation were excluded.

**Outcome measures** Potentially appropriate prescribing of antihypertensives, renin-angiotensin-aldosterone system (RAAS) inhibitors, statins, phosphate binders and potentially inappropriate prescribing according to prevailing guidelines was assessed using prescribing quality indicators. $X^2$ or Fisher's exact tests were used to test for differences in prescribing quality.

**Results** RAAS inhibitors alone or in combination with diuretics (57% or 52%, respectively) and statins (42%) were prescribed less often than phosphate binders (72%) or antihypertensives (94%) when indicated. Active vitamin D was relatively often prescribed when potentially not indicated (19%). Patients with high CKD stages were less likely to receive RAAS inhibitors but more likely to receive statins when indicated than stage 3 CKD patients. They also received more active vitamin D and erythropoietin-stimulating agents when potentially not indicated.

**Conclusions** Priority areas for improvement of prescribing in CKD outpatients include potential underprescribing of RAAS inhibitors and statins, and potential overprescribing of active vitamin D. CKD stage should be taken into account when assessing prescribing quality.

## INTRODUCTION

Assessing quality of care in chronic kidney disease (CKD) patients is important for identifying areas for improvement. Several recent

### Strengths and limitations of this study

► Quality of prescribing was assessed with validated quality indicators measuring both potential appropriate and inappropriate prescribing taking individual patient characteristics into account.
► Quality of prescribing was compared between patients with different CKD stages and from different outpatient clinics.
► Quality of prescribing was assessed in a cross-sectional manner, which disregards longitudinal aspects of care.
► There were differences in patient population and data collection between the three outpatient clinics.

studies have shown that detection of CKD, monitoring of disease progression and metabolic parameters and achievement of risk factor target levels are suboptimal in CKD care.[1–4] Three of these studies showed that prescribing of selected medication treatment may also be suboptimal, for example, showing potential underprescribing of renin-angiotensin-aldosterone system (RAAS) inhibitors and statins, and overprescribing of non-steroidal anti-inflammatory drugs (NSAIDs) in primary care patients with CKD. In addition, one study showed an increasing trend in prescribing of RAAS inhibitors and statins in secondary CKD patients.[5] Not much is known about differences in prescribing quality in CKD patients between healthcare organisations.

Recently, our research group has developed a set of prescribing quality indicators (PQIs) for assessing the prescribing quality in patients with CKD according to clinical guideline recommendations, which has been validated in a primary care population.[6] The set is intended also for secondary care, and includes several indicators that are specifically relevant for patients with higher CKD stages. The PQIs give

new insights in quality of prescribing in CKD patients, since they assess a broad range of prescribing issues at patient level and incorporate patient characteristics that are relevant for treatment decisions. Previously it was found that with increasing CKD stages prescribing of RAAS inhibitors and NSAIDs decreased, while prescribing of phosphate binders, vitamin D and erythropoietin-stimulating agents (ESA) increased.[1] However, this was based on the number of prescriptions regardless of whether the medication was indicated for the included patients. The aim of this study is to assess prescribing quality in secondary care patients with CKD stages 3a–5 using PQIs that take relevant patient characteristics into account. Differences in prescribing quality between patients with different CKD stages were evaluated. In a secondary analysis, also differences between different outpatient clinics in the Netherlands were explored.

## METHODS

This was a retrospective cross-sectional study assessing the prescribing quality between March 2015 and August 2016 in the Netherlands in two university nephrology outpatient clinics A and B, and one non-university nephrology outpatient clinic C. Included were patients with CKD stages 3a–5 based on estimated glomerular filtration rate (eGFR), that is stage 3a was defined as an eGFR $\geq 45$ and $<60 \, mL/min/1.73 \, m^2$, stage 3b as an eGFR $\geq 30$ and $<45 \, mL/min/1.73 \, m^2$, stage 4 as an eGFR $\geq 15$ and $<30 \, mL/min/1.73 \, m^2$ and stage 5 as an eGFR $<15 \, mL/min/1.73 \, m^2$.[7] Patients who received dialysis or renal transplantation were excluded from the study.

The medical ethical committee of the University Medical Center Groningen ascertained that this study using anonymised medical record data does not fall under the Medical Research Involving Human Subjects Act.

### Clinics and setting

Clinic A and clinic B are academic hospitals, which provided data from their general nephrology outpatient clinic and their predialysis outpatient clinic. Clinic C is a nonacademic, public hospital, which provided data from the general nephrology outpatient clinic. Twelve months of consecutive data were collected from 1 June 2015 to 31 May 2016 for clinic A, from 1 March 2015 to 29 February 2016 for clinic B and 1 September 2015 to 31 August 2016 for clinic C. In all three clinics, the included CKD patients commonly visit the outpatient clinics 2–4 times per year depending on the progression of their disease. At these visits, all medication can be reviewed and changed based on a medication list which is available for the nephrologist. In all clinics, the medication included in this study is usually prescribed by the nephrologist or nephrologist in training, although other specialists or the general practitioner may also prescribe medication during the year. Visits for the nephrology and the predialysis outpatient clinics are conducted in a similar way. All patients collect their medication at community pharmacies, where medication reviews may be conducted regardless of the outpatient clinic they visit. All patients have similar access to prescribed medication.

### Prescribing quality

A previously developed set of patient-oriented PQIs was used for the assessment of prescribing quality of patients with CKD not undergoing renal replacement therapy.[6] This set was based on clinical guideline recommendations. The PQIs intend to provide insight in prescribing behaviour of physicians with regard to antihypertensives, RAAS inhibitors, statins and phosphate binders when recommended (potentially appropriate prescribing) as well as prescribing of dual RAAS blockade, active vitamin D, ESA, NSAIDs, metformin and digoxin when considered not needed or unsafe (potentially inappropriate prescribing) (table 1). To specify the indicators to specific needs of patients, most indicators focus on a subgroup of the population selected based on kidney function, risk factor levels and/or age as described in the clinical guidelines or added by the expert panel in the development phase. Since there will always be individual cases for whom this is not the case, we speak of 'potentially' appropriate (or inappropriate) prescribing. Antihypertensives include diuretics, beta blocking agents, calcium channel blockers, agents acting on the RAAS system and other antihypertensives such as centrally acting agents. Diuretics include all types of diuretics, for example low-ceiling and high-ceiling diuretics and potassium-sparing agents. RAAS inhibitors include angiotensin-converting-enzyme inhibitors and angiotensin-receptor blockers. In the Netherlands, this medication is reimbursed by the health insurance after the patient's annual deductible excess has been exceeded. This is the same for all three clinics.

### Data collection

For each patient with at least one visit to a nephrologist within the study period, blood pressure measurement, laboratory measurements and prescription data of the most recent visit were extracted from the medical records, either by manual (clinic A) or computerised (clinics B and C) extraction routines. Age was determined on the visit day. For some patients, the visit date was unknown, in which case the most recent date of the eGFR assessment was used as a proxy for the visit date. The eGFR was calculated from serum creatinine using the modification of diet in renal disease (MDRD) formula.[8] If serum creatinine was not available, the reported eGFR calculated with the MDRD formula was used. Proteinuria was defined as more than 0.5 g of protein in 24 hours or per litre urine, depending on availability.

### Patient and public involvement

Patients or public were not directly involved in this study. Three patient representatives were, however, involved in the development phase of the PQIs applied in this study to ascertain that all relevant topics from the patient perspective were included.[6]

**Table 1** Indicator definitions for the PQIs

| Nr. | Indicator definition |
|---|---|
| **Appropriate prescribing** | |
| 1 | The percentage of patients between 18 and 80 years with CKD stages 4–5 and hypertension* that is prescribed antihypertensives unless undesirable because of low diastolic blood pressure (<70 mm Hg) |
| 2 | The percentage of patients between 18 and 80 years with CKD stages 3–5 and proteinuria† that is prescribed an ACE-i or ARB |
| 3 | The percentage of patients between 18 and 80 years with CKD stages 3–5, micro-albuminuria‡ and diabetes§ that is prescribed an ACE-I or ARB |
| 4 | The percentage of patients between 18 and 80 years with CKD stages 3–5 and proteinuria† treated with multiple antihypertensives that is prescribed a combination of an ACE-i or ARB and a diuretic |
| 5 | The percentage of patients between 18 and 80 years with CKD stages 3–5, micro-albuminuria‡ and diabetes§ treated with multiple antihypertensives that is prescribed a combination of an ACE-I or ARB and a diuretic |
| 6 | The percentage of patients between 50 and 65 years with CKD stages 3–5 that is prescribed a statin |
| 7 | The percentage of patients between 18 and 80 years with CKD stages 3–5 and an elevated phosphate level (>1.49 mmol/L) that is prescribed a phosphate binder |
| 8 | The percentage of patients between 18 and 80 years with CKD stages 3–5 treated with phosphate binders and with an elevated calcium level (>2.54 mmol/L) that is prescribed a non-calcium-containing phosphate binder |
| 9 | The percentage of patients between 18 and 80 years with CKD stages 3–5 treated with phosphate binders and with a low calcium level (<2.10 mmol/L) that is prescribed a calcium-containing phosphate binder |
| **Inappropriate prescribing** | |
| 10 | The percentage of patients 18 years or older with CKD stages 3–5 treated with RAAS inhibitors that is prescribed at least two RAAS inhibitors simultaneously (dual RAAS blockade) |
| 11 | The percentage of patients 18 years or older with CKD stages 3–5 and an elevated calcium level (>2.54 mmol/L) that is prescribed active vitamin D |
| 12 | The percentage of patients 18 years or older with CKD stages 3–5 and a normal haemoglobin level (≥7.5 mmol/L) that is prescribed an ESA |
| 13 | The percentage of patients 18 years or older with eGFR<30 mL/min/1.73 m$^2$ that is prescribed an NSAID |
| 14 | The percentage of patients 18 years or older with eGFR<30 mL/min/1.73 m$^2$ and diabetes§ that is prescribed metformin |
| 15 | The percentage of patients 18 years or older with eGFR<50 mL/min/1.73 m$^2$ treated with digoxin that is prescribed high-dose digoxin (>0.125 mg/day) |
| 16 | The percentage of patients 18 years or older with CKD stages 3–5 that is prescribed a combination of NSAIDs, RAAS inhibitors and diuretics |

*Hypertension is defined as having a systolic blood pressure >140 mm Hg or being prescribed antihypertensives.
†Proteinuria is defined as >0.5 g protein per 24 hours or l urine or albumin/creatinine ratio ≥30 mg/mmol.
‡Micro-albuminuria is defined as albumin/creatinine ratio ≥3.0 mg/mmol and <30 mg/mmol.
§Diabetes is defined as either the diagnosis for diabetes or being prescribed with glucose lowering drugs.
ACE-i, angiotensin-converting-enzyme inhibitor; ARB, angiotensin-receptor blocker; CKD, chronic kidney disease; eGFR, estimated glomerular filtration rate; ESA, erythropoietin-stimulating agent; NSAIDs, non-steroidal anti-inflammatory drugs; PQI, prescribing quality indicator; RAAS, renin-angiotensin-aldosterone system.

## Statistical analysis

Means with SD are reported for normally distributed continuous variables, medians with interquartile ranges for non-normally distributed variables and percentages for categorical variables. The PQI scores are presented as percentages with 95% CI. $X^2$ or Fisher's exact tests, in case of cell frequencies below 5, were used to test for differences in prescribing quality across different CKD stages and different clinics. P-values<0.05 were considered statistically significant. When comparing individual PQIs between CKD stages or clinics, Bonferroni correction for multiple testing was applied. Analyses were conducted using Stata V.14.2 special edition (Stata Corporation).

## RESULTS

In total, 3132 patients with CKD stage 3a-5 were included in this study. Included patients were on average 68 years (SD: 14) old, 56% were males, the median eGFR was 35 mL/min/1.73 m$^2$ (IQR: 24–46) and 16% had diabetes. Patients with higher CKD stages more often had blood pressure and laboratory measurements (table 2).

### Overall prescribing quality

Two PQIs focusing on prescribing of RAAS inhibitors in patients with micro-albuminuria and diabetes were not operational valid in this population because of the limited availability of albumin/creatinine ratios (indicators 3

**Table 2** Patient characteristics for the whole population and separate per CKD stage

| | Overall (n=3132) | | CKD 3a (n=843) | | CKD 3b (n=1125) | | CKD 4 (n=862) | | CKD 5 (n=302) | | Chi² |
|---|---|---|---|---|---|---|---|---|---|---|---|
| | N (%) | Mean (±SD) | N (%) | Mean (±SD) | N (%) | Mean (±SD) | N (%) | Mean (±SD) | N (%) | Mean (±SD) | p-value |
| Age (years) | | 67.7 (±14.1) | | 63.1 (±14.2) | | 68.9 (±13.2) | | 70.2 (±13.8) | | 69.2 (±14.8) | |
| <50 years | 363 (11.6) | | 151 (17.9) | | 113 (10.0) | | 70 (8.1) | | 29 (9.6) | | |
| 50–80 years | 2149 (68.6) | | 604 (71.7) | | 793 (70.5) | | 556 (64.5) | | 196 (64.9) | | |
| >=80 years | 620 (19.8) | | 88 (10.4) | | 219 (19.5) | | 236 (27.4) | | 77 (25.5) | | |
| Gender (males) | 1738 (55.5) | | 456 (54.1) | | 614 (54.6) | | 488 (56.6) | | 180 (59.6) | | 0.314 |
| Diabetes (yes) | 485 (15.5) | | 96 (11.4) | | 166 (14.8) | | 165 (19.1) | | 58 (19.2) | | <0.001 |
| eGFR (MDRD) (ml/min/1.73m²) | 3132 (100) | 35 [24–46]* | 843 (100) | 52.2 (±4.3) | 1125 (100) | 37.3 (±4.2) | 862 (100) | 23.1 (±4.3) | 302 (100) | 11.1 (±2.6) | <0.001 |
| SBP (mmHg) | 2511 (80.2) | 132.0 (±18.8) | 601 (71.3) | 129.4 (±17.3) | 910 (80.9) | 130.7 (±18.6) | 735 (85.3) | 133.3 (±19.4) | 265 (87.7) | 139.1 (±19.2) | <0.001 |
| Elevated SBP (>140mmHg) | 700 (22.3) | 155.3 (±12.4) | 119 (14.1) | 155.3 (±12.3) | 239 (21.2) | 154.3 (±12.5) | 233 (27.0) | 155.4 (±12.2) | 109 (36.1) | 157.1 (±12.6) | <0.001 |
| DBP (mmHg) | 2511 (80.2) | 75.1 (±11.2) | 601 (71.3) | 76.7 (±10.6) | 910 (80.9) | 74.9 (±11.3) | 735 (85.3) | 74.6 (±11.2) | 265 (87.7) | 74.0 (±12.1) | <0.001 |
| Low DBP (<70mmHg) | 704 (22.5) | 61.6 (±5.8) | 130 (15.4) | 62.1 (±5.5) | 282 (25.1) | 62.1 (±5.4) | 214 (24.8) | 61.3 (±6.1) | 78 (25.8) | 59.4 (±6.5) | <0.001 |
| Total protein (g/24h urine) | 1314 (42.0) | 0.4 (0.1–1.3)* | 278 (33.0) | 0.3 (0.1–0.8)* | 426 (37.9) | 0.2 (0.1–0.8)* | 403 (46.8) | 0.4 (0.2–1.3)* | 207 (68.5) | 1.3 (0.5–2.8)* | <0.001 |
| Total protein (g/L urine) | 2328 (74.3) | 0.2 (0.1–0.6)* | 552 (65.5) | 0.1 (0.1–0.3)* | 841 (74.8) | 0.2 (0.1–0.4)* | 687 (79.7) | 0.3 (0.1–0.6)* | 248 (82.1) | 0.8 (0.3–1.7)* | <0.001 |
| Proteinuria (>0.5 g/24h or l urine) | 810 (25.9) | | 140 (16.6) | | 224 (19.9) | | 268 (31.1) | | 178 (58.9) | | |
| Phosphate (mmol/L) | 2606 (83.2) | 1.08 (±0.29) | 539 (63.9) | 0.96 (±0.20) | 959 (85.2) | 1.00 (±0.20) | 815 (94.5) | 1.10 (±0.25) | 293 (97.0) | 1.48 (±0.39) | <0.001 |
| Elevated phosphate (>1.49mmol/L) | 172 (5.5) | 1.80 (±0.34) | 3 (0.4) | 1.69 (±0.22) | 17 (1.5) | 1.65 (±0.20) | 41 (4.8) | 1.72 (±0.32) | 111 (36.8) | 1.85 (±0.36) | <0.001 |
| Calcium (mmol/L) | 2734 (87.3) | 2.36 (±0.14) | 616 (73.1) | 2.38 (±0.11) | 998 (88.7) | 2.37 (±0.13) | 822 (95.4) | 2.35 (±0.15) | 298 (98.7) | 2.30 (±0.16) | <0.001 |
| Elevated calcium (>2.54mmol/L) | 163 (5.2) | 2.62 (±0.08) | 40 (4.7) | 2.61 (±0.08) | 61 (5.4) | 2.62 (±0.09) | 46 (5.3) | 2.62 (±0.06) | 16 (5.3) | 2.64 (±0.08) | <0.001 |
| Haemoglobin (mmol/L) | 3024 (96.6) | 8.0 (±1.1) | 771 (91.5) | 8.5 (±1.1) | 1095 (97.3) | 8.2 (±1.0) | 857 (99.4) | 7.7 (±1.1) | 301 (99.7) | 7.0 (±0.9) | <0.001 |
| Low haemoglobin level (<7.5mmol/L) | 933 (29.8) | 6.7 (±0.6) | 113 (13.4) | 6.8 (±0.6) | 264 (23.5) | 6.8 (±0.5) | 346 (40.1) | 6.7 (±0.6) | 210 (69.5) | 6.6 (±0.7) | <0.001 |
| Clinic | | | | | | | | | | | <0.001 |
| A | 569 (18.2) | | 92 (10.9) | | 166 (14.8) | | 183 (21.2) | | 128 (42.4) | | |
| B | 845 (27.0) | | 255 (30.3) | | 295 (26.2) | | 219 (25.4) | | 76 (25.2) | | |
| C | 1718 (54.9) | | 496 (58.8) | | 664 (59.0) | | 460 (53.4) | | 98 (32.5) | | |

*Median with IQR.

CKD, chronic kidney disease; clinics A and B: university nephrology outpatient clinics; clinic C: non-university nephrology outpatient clinic ; DBP, diastolic blood pressure; eGFR, estimated glomerular filtration rate; MDRD, modification of diet renal disease; SBP, systolic blood pressure.

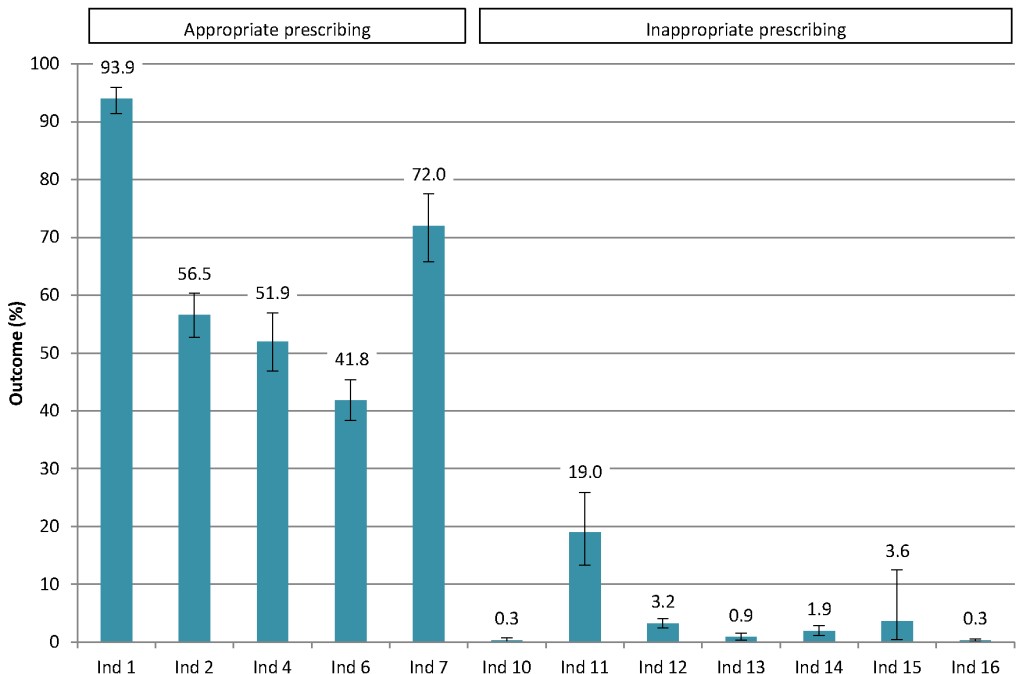

**Figure 1** Overall prescribing quality assessed with five prescribing quality indicators (PQIs) for appropriate prescribing (ind 1–7) and seven PQIs for potential inappropriate prescribing (ind 10–16). Ind 1: patients with hypertension prescribed antihypertensives; ind 2: patients with albuminuria prescribed renin-angiotensin-aldosterone system (RAAS) inhibitors; ind 4: patients on multiple antihypertensives prescribed a combination of RAAS inhibitors and diuretics; ind 6: patients aged 50–65 years prescribed statins; ind 7: patients with high phosphate levels prescribed phosphate binders; ind 10: patients prescribed dual RAAS blockade; ind 11: patients with high calcium levels prescribed active vitamin D; ind 12: patients with normal haemoglobin levels prescribed erythropoietin-stimulating agents; ind 13: patients with an estimated glomerular filtration rate (eGFR) lower than $30\,mL/min/1.73\,m^2$ prescribed high-dose non-steroidal anti-inflammatory drugs (NSAIDs); ind 14: patients with diabetes and an eGFR lower than $30\,mL/min/1.73\,m^2$ prescribed metformin; ind 15: patients with an eGFR lower than $50\,mL/min/1.73\,m^2$ prescribed high-dose digoxin; ind 16: patients prescribed a combination of NSAIDs, RAAS inhibitors and diuretics. 95% CIs were calculated based on included number of patients in the denominator of each indicator.

and 5). Furthermore, two PQIs focusing on prescribing of non-calcium-containing and calcium-containing phosphate binders in patients with high and low calcium levels, respectively, were not operational valid because of the low inclusion of eligible patients (indicators 8 and 9). These four PQIs were excluded from further analysis.

Potentially appropriate prescribing rates varied from 94% of patients receiving antihypertensives, 57% and 52% receiving RAAS inhibitors alone or in combination with a diuretic, 42% receiving statins and 72% receiving phosphate binders when indicated according to the guideline (figure 1). Potentially inappropriate prescribing rates varied from 19% of patients receiving active vitamin D, 3% receiving ESA, 1% receiving NSAIDs, 3% receiving metformin and 4% receiving high-dose digoxin when this was possibly not needed or unsafe.

### Prescribing quality across chronic kidney disease stages
Potential appropriate prescribing of RAAS inhibitors alone occurred significantly less in patients with CKD stage 5 compared with all other stages, which was also true for the combination of RAAS inhibitors and diuretics (figure 2). Patients with stage 3a were less likely to receive recommended treatment with statins than patients with stage 4 or 5. Similarly, patients with stage 3b were less likely

to receive statins compared with patients with stage 4. Potential inappropriate prescribing of active vitamin D in patients with elevated calcium occurred significantly less in patients with stages 3a and 3b compared with patients with stages 4 and 5. This was also the case for potentially inappropriate prescribing of ESA. Finally, potentially inappropriate prescribing of metformin in patients with an eGFR $<30\,mL/min/1.73\,m^2$ was significantly lower for stage 5 compared with stage 4.

### Prescribing quality across nephrology outpatient clinics
Patients visiting the university outpatient clinics A and B were on average younger (63 years (SD: 15) and 65 years (SD: 15)) compared with those visiting the non-university outpatient clinic C (71 years (SD: 13)). Furthermore, patients visiting clinic A more often had CKD stage 4 or 5 compared with patients from clinics B and C (table 2). The diabetes prevalence was higher at clinic A (26%) compared with clinic B (19%) and clinic C (10%) (online supplementary table 1).

Significant differences were seen between clinic A and clinic C in potentially appropriate prescribing of antihypertensives, RAAS inhibitors alone, statins and in potentially inappropriate prescribing of metformin as well as the combination of NSAIDs, RAAS inhibitors

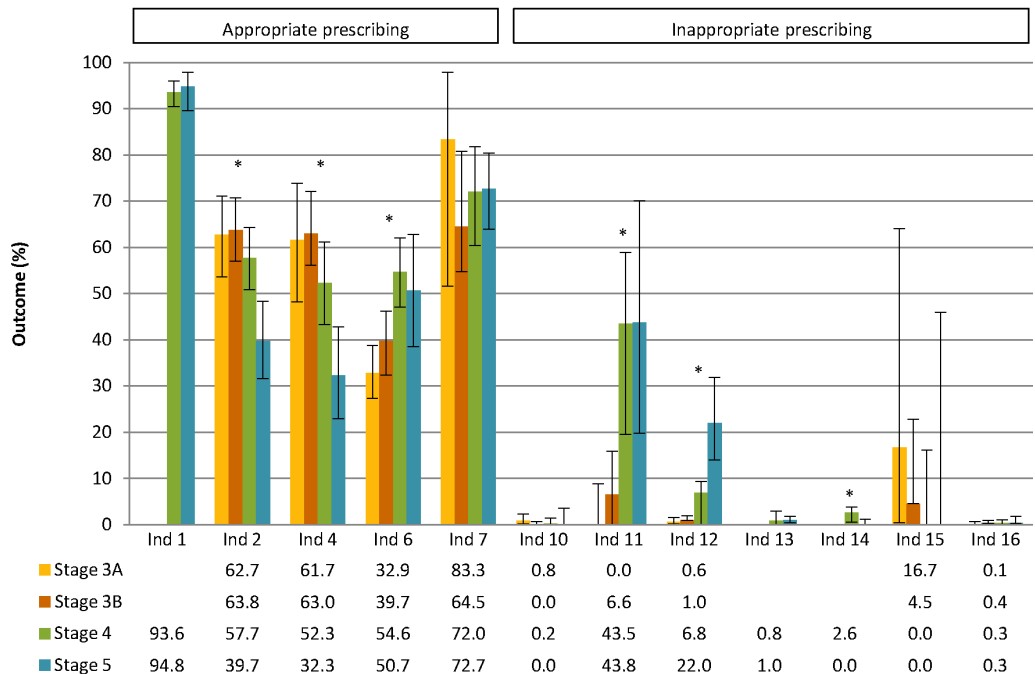

| | Ind 1 | Ind 2 | Ind 4 | Ind 6 | Ind 7 | Ind 10 | Ind 11 | Ind 12 | Ind 13 | Ind 14 | Ind 15 | Ind 16 |
|---|---|---|---|---|---|---|---|---|---|---|---|---|
| Stage 3A | | 62.7 | 61.7 | 32.9 | 83.3 | 0.8 | 0.0 | 0.6 | | | 16.7 | 0.1 |
| Stage 3B | | 63.8 | 63.0 | 39.7 | 64.5 | 0.0 | 6.6 | 1.0 | | | 4.5 | 0.4 |
| Stage 4 | 93.6 | 57.7 | 52.3 | 54.6 | 72.0 | 0.2 | 43.5 | 6.8 | 0.8 | 2.6 | 0.0 | 0.3 |
| Stage 5 | 94.8 | 39.7 | 32.3 | 50.7 | 72.7 | 0.0 | 43.8 | 22.0 | 1.0 | 0.0 | 0.0 | 0.3 |

**Figure 2** Prescribing quality across different chronic kidney disease (CKD) stages (3a–5) assessed with five PQIs for appropriate prescribing (ind 1–7) and seven PQIs for potential inappropriate prescribing (ind 10–16). Ind 1: patients with hypertension prescribed antihypertensives; ind 2: patients with albuminuria prescribed renin-angiotensin-aldosterone system (RAAS) inhibitors; ind 4: patients on multiple antihypertensives prescribed a combination of RAAS inhibitors and diuretics; ind 6: patients aged 50 to 65 years prescribed statins; ind 7: patients with high phosphate levels prescribed phosphate binders; ind 10: patients prescribed dual RAAS blockade; ind 11: patients with high calcium levels prescribed active vitamin D; ind 12: patients with normal haemoglobin levels prescribed erythropoietin-stimulating agents; ind 13: patients with an estimated glomerular filtration rate (eGFR) lower than 30 mL/min/1.73 m$^2$ prescribed high-dose non-steroidal anti-inflammatory drugs (NSAIDs); ind 14: patients with diabetes and an eGFR lower than 30 mL/min/1.73 m$^2$ prescribed metformin; ind 15: patients with an eGFR lower than 50 mL/min/1.73 m$^2$ prescribed high-dose digoxin; ind 16: patients prescribed a combination of NSAIDs, RAAS inhibitors and diuretics. 95% CIs were calculated based on included number of patients in the denominator of each indicator. *Significant difference between two or more CKD stages using $X^2$ or Fisher's exact test with Bonferroni correction for multiple testing.

and diuretics (figure 3). Furthermore, significantly more potentially appropriate prescribing of phosphate binders was seen in clinic A compared with clinic B.

In the analyses per CKD stage (online supplementary tables 2 and 3), similar differences were found between the clinics. In patients with stage 3a CKD, potentially appropriate prescribing of RAAS inhibitors combined with diuretics occurred the least in clinic B. In patients with stage 3b CKD, potential appropriate prescribing of RAAS inhibitors alone or combined with diuretics occurred the most in clinic A. Patients with stage 4 CKD were significantly more likely in clinic A compared with clinic C to receive antihypertensives and RAAS inhibitors alone. Also, patients with CKD stage 5 were more likely in clinic A compared with clinic C to receive phosphate binders when indicated.

## DISCUSSION

This is the first study to assess the prescribing quality in secondary care CKD patients using a broad set of patient-oriented PQIs and comparing patients with different CKD stages and from different outpatient clinics. The results show that the prescribing quality

seems to vary between therapeutic areas. RAAS inhibitors and statins were prescribed in less than 60% of the patients for whom this is potentially indicated, whereas prescribing rates for antihypertensives and phosphate binders when potentially indicated were much higher. Active vitamin D was prescribed in almost one fifth of all patients while considered potentially not needed or unsafe. The prescribing levels also varied across different CKD stages, with decreasing prescribing of RAAS inhibitors, increasing prescribing of statins and increasing prescribing of active vitamin D and ESA with higher CKD stages. Finally, significant differences were observed in prescribing between the different outpatient clinics, also after stratification for CKD stage.

Previous studies looking at the overall volume of prescribing suggested that there was underprescribing of RAAS inhibitors and statins[1 5] and overprescribing of NSAIDs[1] in patients with CKD. In addition, it was shown that patients with higher CKD stages receive more treatment with antihypertensives, phosphate binders, vitamin D and ESA than patients with lower CKD stages.[1 9] On the other hand, RAAS inhibitors and NSAIDs were less prescribed with increasing CKD stages.[1] These studies,

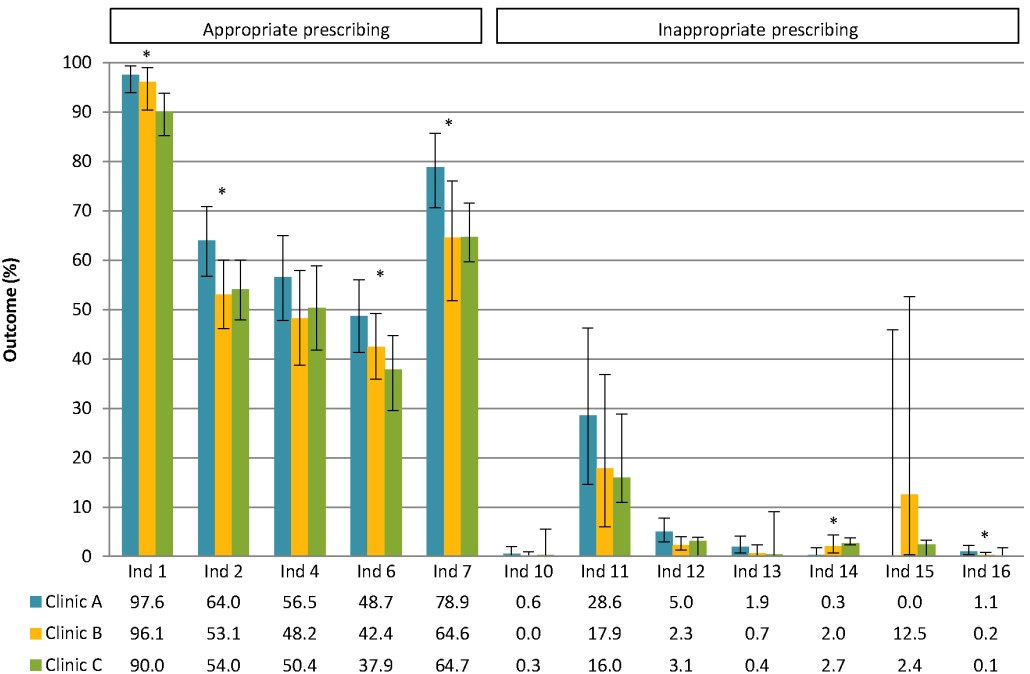

**Figure 3** Prescribing quality across different outpatient clinics. Clinics A and B: university nephrology outpatient clinics; clinic C: non-university nephrology outpatient clinic. Ind 1: patients with hypertension prescribed antihypertensives; ind 2: patients with albuminuria prescribed renin-angiotensin-aldosterone system (RAAS) inhibitors; ind 4: patients on multiple antihypertensives prescribed a combination of RAAS inhibitors and diuretics; ind 6: patients aged 50–65 years prescribed statins; ind 7: patients with high phosphate levels prescribed phosphate binders; ind 10: patients prescribed dual RAAS blockade; ind 11: patients with high calcium levels prescribed active vitamin D; ind 12: patients with normal haemoglobin levels prescribed erythropoietin-stimulating agents; ind 13: patients with an estimated glomerular filtration rate (eGFR) lower than 30 mL/min/1.73 m$^2$ prescribed high-dose non-steroidal anti-inflammatory drugs (NSAIDs); ind 14: patients with diabetes and an eGFR lower than 30 mL/min/1.73 m$^2$ prescribed metformin; ind 15: patients with an eGFR lower than 50 mL/min/1.73 m$^2$ prescribed high-dose digoxin; ind 16: patients prescribed a combination of NSAIDs, RAAS inhibitors and diuretics. 95% CIs were calculated based on included number of patients in the denominator of each indicator. *Significant difference between two or all outpatient clinics using $X^2$ or Fisher's exact test with Bonferroni correction for multiple testing.

however, did not take the specific indications for treatment into account. Our study using validated PQIs, which assess prescribing in patients for whom this is indicated, confirmed that potential underprescribing of RAAS inhibitors and statins may be areas for possible improvement in CKD care. This was also found in a recent study among patients with stages 3–4 CKD in Canada.[3] In some patients with CKD stage 5 who are in preparation of dialysis or transplantation, RAAS inhibitors may be deliberately stopped to retain the residual kidney function.[10] Lower statin prescribing rates in patients with lower CKD stages suggest that prescribers may be less aware or convinced of the need to prescribe statins in these patients. A similar pattern of less statin prescribing in patients with a higher eGFR compared with lower eGFR was observed for the elderly primary care patients in Canada.[11] Regarding potentially inappropriate prescribing, the Canadian primary care study observed a relatively high prescribing rate of NSAIDs (16%) and low prescribing rate of dual RAAS blockade (3%). Our study showed that potentially unsafe prescribing of both NSAIDs and dual RAAS blockade was uncommon in secondary care patients managed in the Netherlands.

Although the applied PQIs are patient-oriented, taking relevant patient characteristics into account, they reflect general guideline recommendations, and therefore a perfect score is never pursued. There can be valid reasons to refrain from prescribing according to guideline recommendations in certain patients. Valid reasons include lack of response to certain drugs, drug intolerances or patient preferences for or against certain treatment. For example, hyperkalaemia could be a reason to stop treatment with RAAS inhibitors. The assessment using PQIs is therefore useful to provide insight and monitor the quality of care at population level and not to assess the quality of treatment for each individual patient. Furthermore, it has been argued that patient case-mix including difference in aspects, such as age or comorbidities, may explain differences in quality scores.[12] However, these may not necessarily be valid reasons for not complying with guideline recommendations. When developing the PQIs, such differences are to some extent included in the indicator definitions (eg, age limits), thereby ensuring that the treatments are in general either recommended or inappropriate in the patients included in the indicator. Furthermore, a recent review showed that unjustified

case-mix corrections can mask actual differences in quality of care.[13] Therefore, no case-mix adjustment was made when applying the PQIs.

This study assessed the prescribing quality in a cross-sectional manner, since the PQIs were defined as cross-sectional measures. This may lead to including patients who reached abnormal risk factor levels for the first time. In some cases, the healthcare provider may decide to postpone the start of treatment until a next measurement to make sure that the abnormal risk factor level persists. This also holds for discontinuation of active vitamin D in patients with elevated calcium levels and ESA in patients with normal haemoglobin levels. Furthermore, it is possible that the laboratory results became available after the visit. Therefore, the healthcare provider may not have been aware at the time of the visit that they should start or discontinue medication. In the diabetes field, it has been proposed that indicators using multiple time points may give a more accurate assessment of the prescribing quality.[14 15] Such indicators assess whether the healthcare providers start or intensify treatment when patients do not return to normal risk factor levels.

Differences between clinics may in part be due to differences in the underlying patient population such as age and comorbidity. All PQIs focusing on recommended treatment with antihypertensives, RAAS inhibitors, statins and phosphate binders, however, have an age limit which excludes the older, more frail patients. Diabetes prevalence was higher in clinic A, which may have affected prescribing. Other studies indicate that CKD patients with diabetes receive better quality of care in general,[16] and have higher prescription rates of RAAS inhibitors and statins.[3] There were also differences in data collection methods between the clinics. Although we were able to extract and combine the data in order to make comparisons possible, there were some differences in availability of measurement values. Furthermore, data from the blood pressure and laboratory measurements were sometimes missing (table 2), with the highest rate of missingness for clinic B (online supplementary table 2). This could have influenced the outcome of the PQIs, since patients with unknown values were not included in the PQIs. We can only speculate why these values were missing, and how this may have influenced the assessments. It could be that the blood pressure and laboratory measurements were not performed, not recorded or lost during data extraction. Our aim was not to explain differences in prescribing. To understand the causes underlying prescribing variation, future studies should look at the influence of patient, prescriber and organisation characteristics.

In conclusion, using a novel set of PQIs assessing prescribing quality with patient level data we successfully identified several areas for potential improvement. This included potential underprescribing of RAAS inhibitors and statins and potential overprescribing of active vitamin D in secondary care patients with CKD stages 3–5. This information can support clinicians to identify the patients that are truly in need of improved treatment. We observed differences in prescribing quality between the CKD stages and between the outpatient clinics. We conclude that monitoring of the prescribing quality with PQIs in secondary care and stratification on CKD stage can be used to identify priority areas for quality improvement initiatives.

**Acknowledgements** We would like to thank the following people for the data collection: Judith Bos (Clinic A), Thijme Last (Clinic B) and Bert Dikkeschei and Erwin Booij (Clinic C).

**Contributors** KPJS: research idea; study design; planning; data acquisition; data analysis and interpretation; GS: research idea; study design; data analysis and interpretation; supervision or mentorship; FJvl and FW: data acquisition; data interpretation; HJGB, GJN and PD: research idea; study design; data acquisition; data analysis and interpretation; supervision or mentorship. All authors contributed to important intellectual content during manuscript drafting or revisions. All authors approved the manuscript and this submission.

**Funding** This work was supported by ZonMW, the Netherlands Organisation for Health Research and Development, grant number 836021013. The funder had no role in study design, data collection and analysis, decision to publish or preparation of the manuscript.

**Competing interests** None declared.

**Patient consent for publication** Not required.

**Provenance and peer review** Not commissioned; externally peer reviewed.

**Data sharing statement** Clinical data as collected from the outpatient clinics cannot be made publicly available according to Dutch privacy regulations. Data sets are only available for data integrity auditing procedures.

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
