## [Reviewer comments · BMJ Open]

ARTICLE DETAILS

TITLE (PROVISIONAL)	Prescribing quality in secondary care patients with different stages of chronic kidney disease: a retrospective study in the Netherlands
AUTHORS	Smits, Kirsten; Sidorenkov, Grigory; van Ittersum, Frans; Waanders, Femke; Bilo, Henk; Navis, GJ; Denig, Petra

VERSION 1 - REVIEW

REVIEWER	Gang Xu Renal unit University Hospitals of Leicester England
REVIEW RETURNED	20-Sep-2018

GENERAL COMMENTS	Very interesting study, the results are clearly presented, however I think the conclusions need to reflect the limitations of the study in more detail. • There is a clear selection bias in the different populations of clinic A/B/C, in particular the number of patients with CKD 4/5. If possible the authors should try to describe the nature of how patients are grouped in clinic A/B/C and how the clinics are run. For example is clinic A runs as 'pre dialysis' service, in which case are appointment times longer? Is there access to pre-dialysis nurses?• I can also see no data on serum K+ results, the major risk/complication of RAAS treatment in hyperkalaemia, this data would be very helpful in understanding the results. Medication prescription in all patients is a balance of benefits versus risk, care needs to be taken when trying to understand prescription practice in complex patients such as those with CKD. Raw data often doesn't reflect the decision making process that occurs in clinical consultations, or the importance of patient choice. I think it's hard to concluded that RASS are 'under-prescribed' based on the data presented, equally the authors have suggested Vit D prescriptions may be hindered by delay/lag of receiving serum Ca results. Clearly a better understanding of medication prescription in CKD patients is needed, but I would be careful to suggest even 'potential' under prescribing given the bias in population, lack of some key biochemical data, and no detailed analysis on the process of care that is carried out in CKD/Predialysis clinics. I would also imagine patient input into any future work in this area is
---

	vital, as medication prescription should never done without patient consultation.
--	---

REVIEWER	Mark Brady Department of Renal Medicine Royal Preston Hospital Lancashire Teaching Hospitals NHS Foundation Trust UK
REVIEW RETURNED	07-Oct-2018

GENERAL COMMENTS	This is an interesting and important topic for CKD management. This work will be of interest to all renal departments to reflect upon how they can be confident in their prescribing patterns. The aims and methodology are clear. The phrase appropriate or inappropriate makes sense but it felt as if the authors needed to know more about the patients in their cohort - the reason for not prescribing indicated medications should ideally be clear for each individual but this was absent from this study. This study indicates a gap between recommended prescribing guidance and actual uptake. It does not explain the underlying causes. I wondered if it was within the study data to consider the following to help explain for example:  1. Does sociodemographic background of patients influence appropriate prescribing - baseline characteristics described in study table do not describe this aspect 2. Do all patients (again considering locality & sociodemographics) have same access to prescribed medications once issued by University or non-university hospital in Netherlands? Did frequency of visits make difference? 3. What resources are available in each centre? Such as skill mix of specialists to doctors in training or pharmacy support. Was the level of appropriate prescribing influenced by experience of prescriber? Overall I enjoyed this paper and felt it highlighted a vital issue in caring for CKD patients. It has raised several questions based on findings and I felt this could have been addressed with a few sentences in discussion regarding limitations and confounders or explanations.
--

VERSION 1 – AUTHOR RESPONSE

Reviewer(s)' Comments to Author:

Reviewer: 1

Reviewer Name: Gang Xu

Institution and Country: Renal unit, University Hospitals of Leicester, England. Please state any competing interests or state 'None declared': None declared

Please leave your comments for the authors below: Very interesting study, the results are clearly presented, however I think the conclusions need to reflect the limitations of the study in more detail.

- There is a clear selection bias in the different populations of clinic A/B/C, in particular the number of patients with CKD 4/5. If possible the authors should try to describe the nature of how patients are grouped in clinic A/B/C and how the clinics are run. For example is clinic A runs as 'pre dialysis' service, in which case are appointment times longer? Is there access to pre-dialysis nurses?

We agree with the reviewer that there is likely to be some selection bias in the different clinics. As mentioned in the manuscript, Clinics A and B are university outpatient clinics while Clinic C is a non-university outpatient clinic. As described in the subsection Clinics in the Methods section, the way these clinics are run is in general similar. We have added a sentence to this section to clarify that this also holds for the predialysis outpatient clinics. On the other hand, these predialysis patients are by definition a different patient group than 'regular' CKD patients. As such, the focus of treatment of these patients might be different as well. The prescribing quality indicators used in this manuscript, however, are based on recommendations for all CKD patients –including predialysis patients- not undergoing renal replacement therapy. We also added this clarification in the Methods section, subsection 'Prescribing quality': "A previously developed set of patient-oriented PQIs was used for the assessment of prescribing quality of patients with CKD not undergoing renal replacement therapy.

- I can also see no data on serum K⁺ results, the major risk/complication of RAAS treatment in hyperkalaemia, this data would be very helpful in understanding the results.

We thank the reviewer for this comment. Unfortunately, the serum K⁺ lab results for these patients were unavailable for this study. We assessed the treatment at population level, making use of prescribing quality indicators that were previously developed in which serum K⁺ levels were not included. These prescribing quality indicators are derived from clinical guideline recommendations and validated by an expert panel. The indicators do take general patient characteristics into account but do not adjust for individual circumstances that would prevent using the guideline-recommended treatment. This can be seen as a limitation of all prescribing quality indicators, implying that they should only be used as indicators of treatment quality at population level and not as instruments to assess the quality of treatment for each individual. It is assumed that the bias resulting from this is similar across populations. We address this in the discussion (page 12 to 13). Given your comment, we have rephrased this part of the discussion and added the example of hyperkalaemia.

Medication prescription in all patients is a balance of benefits versus risk, care needs to be taken when trying to understand prescription practice in complex patients such as those with CKD. Raw data often doesn't reflect the decision making process that occurs in clinical consultations, or the importance of patient choice. I think it's hard to conclude that RAAS are 'under-prescribed' based on the data presented, equally the authors have suggested Vit D prescriptions may be hindered by delay/lag of receiving serum Ca results.

Clearly a better understanding of medication prescription in CKD patients is needed, but I would be careful to suggest even 'potential' under prescribing given the bias in population, lack of some key biochemical data, and no detailed analysis on the process of care that is carried out in CKD/Predialysis clinics. I would also imagine patient input into any future work in this area is vital, as medication prescription should never be done without patient consultation.

These comments are in line with the previous comment and we agree that the indicators should not be used to draw hard conclusions on under- or overprescribing of medication at the individual patient level. Therefore, these kinds of assessments are usually described as 'potential' under- or overprescribing. This is also explained in the Methods section under 'Prescribing quality'. The value of these kinds of assessments lies in their signal/indicator function. A score of 100% is never pursued in these kind of indicators because there may be valid reasons to deviate from the clinical guidelines, including patient preferences (as mentioned in our discussion). In clinical practice, feedback on these indicators should be followed by more in-depth evaluation of individual cases. Following from your comments, we have emphasized this matter more clearly by using the term potential throughout the

manuscript. Moreover, we have rephrased the conclusion to clarify the meaning and value of our findings for clinical practice.

Patient input is indeed relevant in this area and patient representatives were consulted in the development process of the prescribing quality indicators (See also Methods section, 'Patient and Public involvement') to ensure that aspects of treatment that they considered important were included in the indicators.

Reviewer: 2

Reviewer Name: Mark Brady

Institution and Country: Department of Renal Medicine Royal Preston Hospital, Lancashire Teaching Hospitals NHS Foundation Trust, UK. Please state any competing interests or state 'None declared':
None declared

Please leave your comments for the authors below: This is an interesting and important topic for CKD management. This work will be of interest to all renal departments to reflect upon how they can be confident in their prescribing patterns. The aims and methodology are clear. The phrase appropriate or inappropriate makes sense but it felt as if the authors needed to know more about the patients in their cohort - the reason for not prescribing indicated medications should ideally be clear for each individual but this was absent from this study.

We agree with the reviewer that drawing hard conclusions on appropriate and inappropriate prescribing is not possible using prescribing quality indicators based on general guideline recommendations. Given this comment, and also comments of reviewer 1, we have emphasized that this is about 'potentially' appropriate/inappropriate prescribing throughout the manuscript. We have also rephrased our conclusion accordingly.

This study indicates a gap between recommended prescribing guidance and actual uptake. It does not explain the underlying causes. I wondered if it was within the study data to consider the following to help explain for example:

1. Does sociodemographic background of patients influence appropriate prescribing - baseline characteristics described in study table do not describe this aspect

When trying to understand differences in uptake of prescribing recommendations, more information about both patients and prescribers would be required. Unfortunately our data did not include information on the socioeconomic background of the patients. According to the guideline recommendations, prescribing should not be influenced by socioeconomic status of patients. The socioeconomic status could be seen as a case-mix variable. In the discussion the aspect of case-mix is addressed.

2. Do all patients (again considering locality & sociodemographics) have same access to prescribed medications once issued by University or non-university hospital in Netherlands? Did frequency of visits make difference?

Access to prescribed medication is similar for all patients in the Netherlands. We did not have information on frequency of visits. Although we understand your interest in explaining variations in prescribing, the primary aim of our study was to assess prescribing as recommended by the guidelines. We have added a sentence about access to medication to the Methods subsection Clinics and setting.

3. What resources are available in each centre? Such as skill mix of specialists to doctors in training or pharmacy support. Was the level of appropriate prescribing influenced by experience of prescriber?

As mentioned in reaction to your first remark, when trying to explain differences it would be interesting to have more information about the healthcare providers at the clinics. Given the purpose of our study, we have no information on characteristics of the prescribers. We have added a sentence to the discussion about future research addressing such issues.

As described in the Methods section, subsection Clinics, the provision of care was similar in all clinics, including both nephrologists and nephrologists in training. Patients visiting outpatient clinics collect their medication at community pharmacies, where medication reviews may be conducted in elderly patients regardless of the outpatient clinic they visit. We have added a sentence about this in the subsection Clinics.

Overall I enjoyed this paper and felt it highlighted a vital issue in caring for CKD patients. It has raised several questions based on findings and I felt this could have been addressed with a few sentences in discussion regarding limitations and confounders or explanations.

Thank you for your kind words. We have used your comments to add several sentences in the Methods as well as the Discussion section. Given the primary aim of our study, we did not include any patient background characteristics as 'confounders'. As mentioned in our discussion, unjustified case-mix corrections can mask actual differences in quality of care. We agree very much with you that future studies trying to explain variation in prescribing should include patient, prescriber and also organisation characteristics. As said, we have added this to the discussion.

VERSION 2 – REVIEW

REVIEWER	Gang XU John Walls Renal Unit University Hospitals of Leicester Leicester UK
REVIEW RETURNED	30-Apr-2019

GENERAL COMMENTS	Overall a detailed and interesting article. The authors have clearly a lot of time ensuring the data analysis and results are presented in a readable manner. The results are interesting and demonstrates a need to better understand clinical prescription practice.
--